# MicroRNA156 (miR156) Negatively Impacts Mg-Protoporphyrin IX (Mg-Proto IX) Biosynthesis and Its Plastid-Nucleus Retrograde Signaling in Apple

**DOI:** 10.3390/plants9050653

**Published:** 2020-05-22

**Authors:** Qingbo Zheng, Yakun Chen, Xiaolin Jia, Yi Wang, Ting Wu, Xuefeng Xu, Zhenhai Han, Zhihong Zhang, Xinzhong Zhang

**Affiliations:** 1College of Horticulture, China Agricultural University, Beijing 100193, China; B20163170701@cau.edu.cn (Q.Z.); B20163170693@cau.edu.cn (Y.C.); wangyi@cau.edu.cn (Y.W.); wuting@cau.edu.cn (T.W.); xuefengx@cau.edu.cn (X.X.); rschan@cau.edu.cn (Z.H.); 2Plant Genomics and Molecular Breeding, Henan Agricultural University, Zhengzhou 450002, China; jiaxiaolin@cau.edu.cn; 3Horticulture College, Shenyang Agricultural University, Shenyang 110161, China

**Keywords:** *Malus*, miR156, plastid-nucleus retrograde signaling, tetrapyrrole, vegetative phase change

## Abstract

Plastid-nucleus retrograde signaling (PNRS) play essential roles in regulating nuclear gene expression during plant growth and development. Excessive reactive oxygen species can trigger PNRS. We previously reported that in apple (*Malus domestica* Borkh.) seedlings, the expression of microRNA156 (miR156) was significantly low in the adult phase, which was accompanied by high levels of hydrogen peroxide (H_2_O_2_) accumulation in chloroplasts. However, it was unclear whether adult-phase-specific chloroplast H_2_O_2_ may induce PNRS and affect miR156 expression, or miR156 triggers adult phase PNRS during the ontogenesis. In this paper, we examined the relationship between miR156 levels and six PNRS components in juvenile and adult phase leaves from ‘Zisai Pearl’×‘Red Fuji’ hybrids. We found that PNRS generated by singlet oxygen (^1^O_2_), the photosynthetic redox state, methylerythritol cyclodiphosphate (MEcPP), SAL1-3-phosphoadenosine 5-phosphate (PAP) and WHIRLY1 were not involved. The accumulation of Mg-protoporphyrin IX (Mg-Proto IX), the expression of the synthetic genes *MdGUN5* and *MdGUN6*, and Mg-Proto IX PNRS related nuclear genes increased with ontogenesis. These changes were negatively correlated with miR156 expression. Manipulating Mg-Proto IX synthesis with 5-aminolevulinic acid (ALA) or gabaculine did not affect miR156 expression in vitro shoots. In contrast, modulating miR156 expression via *MdGGT1* or *MdMIR156a6* transgenesis led to changes in Mg-Proto IX contents and the corresponding gene expressions. It was concluded that the Mg-Proto IX PNRS was regulated downstream of miR156 regardless of adult-phase-specific plastid H_2_O_2_ accumulation. The findings may facilitate the understanding of the mechanism of ontogenesis in higher plants.

## 1. Introduction

The vegetative phase change in later diverging land plants is indicative of a loss of juvenility and the acquisition of reproductive competence [1]. During this transition, many morphological, physiological, biochemical, and molecular changes have been documented, including leaf heteroblasty, trichome formation, secondary metabolite accumulation, and gene expression [2,3,4]. The revolutionary discovery of miR156 as the determinant of juvenility unveils the mystery eventually of the vegetative phase change in flowering plants [5]. The effect of high miR156 levels on maintaining juvenility is conserved among diverse plant species. An artificially induced high miR156 expression leads to rejuvenation and a delay in reproductive development [6,7]. A miR156-*SQUAMOSA-PROMOTER BINDING PROTEIN-LIKE (SPL)* regulatory module directly controls transcription of its target genes, such as miR172, MADS-box, and *FLOWERING LOCUS T* (*FT*) [8,9].

The levels of mature miR156 are regulated predominantly by the transcription of its precursor genes [10]. In *Arabidopsis thaliana,* the major precursor genes of miR156 are MIR156A and MIR156C [10]. In the leaf and shoot apex of apple (*Malus domestica* Borkh.), miR156 levels are attributed to *MdMIR156a5* and *MdMIR156a12* transcription [11]. Other factors, such as redox homeostasis, sugar levels, gibberellic acid, and epigenetic regulation, affect the transcription of the miR156 precursor genes [11,12,13,14].

The cellular redox status is determined by the balance of antioxidants and reactive oxygen species (ROS). Redox homeostasis influences cell proliferation, tissue differentiation, organ development, and aging [15]. High doses of ROS can trigger programmed cell death (PCD), but low levels of ROS can act as signaling molecules [16,17]. The most abundant and stable form of cellular ROS is H_2_O_2_. We previously found that H_2_O_2_ levels increase gradually with node number, which is negatively correlated with the expression of miR156 and its precursors [11,18]. Although H_2_O_2_ can migrate across membranes, either by diffusion or to be transported by aquaporins [16,19], an adult-phase-specific high level of H_2_O_2_ was detected in chloroplasts [11]. 

The possible sources of this elevated H_2_O_2_ level in the adult phase could be either increased ROS production or insufficient scavenging. It is known that H_2_O_2_ produced by photosynthetic electron leakage are higher than those in mitochondria [20]. However, the expression of photosynthesis-associated genes does not change during the vegetative phase transition in apple [21]. Moreover, both the content of ascorbate (ASC), an important ROS scavenger, and the ascorbate/dehydroascorbate ratio increase during the phase change in apple [18]. In contrast, the concentration of glutathione (GSH), another major ROS scavenger in the plastid, and glutathione/glutathione disulfide (GSH/GSSG) ratio are significantly lower in adult phase than in juvenile phase [22]. Plastid H_2_O_2_ also perceives stress and developmental cues and triggers PNRS, which regulates nuclear gene transcription [23].

In chloroplasts, the oxidized nicotinamide adenine dinucleotide phosphate (NADP^+^/NADPH) ratio is balanced by the NADPH-thioredoxin reductase C/2-Cys peroxiredoxin (NTRC/2-Cys Prx) system to sustain ROS scavenging [24]. However, it is not known whether the chloroplast H_2_O_2_ quencher NTRC/2-Cys Prx system undergoes a change with age [24]. 

^1^O_2_ is generated by the photosystem II (PS II) reaction center in the chloroplast. It induces two plastid-localized proteins, EXECUTER1 (EX1) and EXECUTER2 (EX2), which transmit PNRS [25]. Plastid β-carotene quenches ^1^O_2_ and is oxidized to form gaseous β-cyclocitral (β-CC), which then traffics across lipid membranes to regulate nuclear gene expression [26]. The expression of several nuclear genes in *Arabidopsis thaliana*, such as *AAA-ATPase, GSTU13, LOX2, OPR3, SCL14, WRKY33, WRKY40, SIB1, AOC4*, are regulated by ^1^O_2_ triggered PNRS [25,27,28,29].

Previously proposed as a “master switch” in retrograde signaling pathways, GENOMES UNCOUPLED1 (GUN1) has been described as a central integrator of the PNRS pathway. GUN1 perceives the redox signal of the photosynthetic electron transfer chain and controls the expression of a large number of nuclear genes [30]. Recently, however, the involvement of GUN1 and the post-translational modification of its targets during retrograde signaling have been questioned [31].

Several secondary metabolites may also serve as PNRS. Tocopherol (vitamin E) in chloroplasts is required for the accumulation of PAP. PAP protects primary miRNAs from being degraded by exoribonucleases and promotes mature miRNA production [32]. MEcPP, a plastid isoprenoid precursor, can migrate to the nucleus under stress conditions. MEcPP binds to the CALMODULIN-BINDING TRANSCRIPTION ACTIVATOR 3 (CAMTA3) transcription factor and activates the transcription of stress responsive genes [33]. WHIRLY1 is a mobile chloroplast transcription factor; WHIRLY1 24-oligomer perceives redox signals and is monomerized. The WHIRLY 1 monomer is recruited to the nucleus and regulates the expression of *NONEXPRESSOR OF PATHOGENESIS-RELATED 1* (*NPR1*) genes [34]. 

Other important PNRS molecules are Fe-protoporphyrin IX (heme) and Mg-Proto IX from the tetrapyrrole biosynthetic pathway. Mg^2+^ ion is inserted into protoporphyrin IX by magnesium chelatase (CHL) to form Mg-Proto IX in chloroplast [35]. However, the accumulation of Mg-Proto IX was also observed in the cytosol [36]. Several nuclear genes containing G-box motif may respond to Mg-Proto IX signal, e.g., *LHCB* genes expression in *Arabidopsis* protoplasts was significantly repressed by addition of Mg-Proto IX [37]. Other nuclear genes, such as *GUN4*, *CBP*, *HY5*, *APRR5*, *ZTL*, *GLK1*, *GLK2*, and *RBCS*, are regulated by Mg-Proto IX [38,39,40]. The Mg-Proto IX synthetic enzyme, Mg-chelatase CHL consists of 80 kDa CHLD, 40 kDa CHLI, and 140 kDa CHLH subunits [41]. CHLH, encoded by the nuclear gene *GUN5*, interacts with WRKY40 in cytosol, thereby affecting the expression of WRKY40-targeted nuclear genes [42,43]. The last reaction of heme biosynthesis in chloroplasts is catalyzed by ferrochelatase (FC). In *Arabidopsis*, heme, catalyzed by FC1 (encoded by *GUN6*), is a PNRS regulating photosynthesis-associated nuclear gene (PhANG) expression [44].

To date, there is no evidence explaining whether adult-phase-specific chloroplast H_2_O_2_ induced PNRS affects miR156 expression, or otherwise miR156 triggers adult phase plastid ROS accumulation and subsequent PNRS. In this current study, to address this question, the interrelations between several PNRS and miR156 expression were investigated in apple hybrids.

## 2. Results

### 2.1. Re-Examination of Age-Related H_2_O_2_ Accumulation and Photosynthetic Activity

We previously reported that higher amounts of H_2_O_2_ accumulated in plastids of adult phase leaves than that of juvenile phase [11]. To exclude the possible effect of differences in physical position and/or light exposure acquisition on leaf H_2_O_2_ content or photosynthesis, the H_2_O_2_ content and photosynthetic properties were re-examined using vegetative propagated trees. Significantly higher H_2_O_2_ content was detected in scion leaves from the adult phase than that from the juvenile phase. This confirmed the age-related H_2_O_2_ accumulation in adult phase (Figure 1A). The mean net photosynthetic rate (NPR) was slightly higher in scion leaves from the adult phase than that from the juvenile phase, but the differences were not statistically significant (Figure 1B). Due to the genetic hyper-heterozygosity and extensive offspring segregation, the H_2_O_2_ and NPR varied among three hybrids also. No substantial changes in minimal/maximal fluorescence and Fv/Fm values were detected (Figure 1C–E). These data indicated that there was no variation in the net photosynthetic rate or photosynthetic electron transport between ontogenetic phases.

### 2.2. Re-Examination of Changes in GSH Levels and the GSH/GSSG Ratio

Using grafted trees, we re-investigated the concentration of GSH, GSSG, GSH+GSSG, and the GSH/GSSG ratio in the juvenile and the adult phase leaves. The GSSG+GSH concentration did not vary significantly, except that was relatively higher in juvenile sample from the 07-07-119 (*p* = 0.07) (Figure 2A). Significantly higher GSSG contents were detected in the adult phase than that in the juvenile phase of 07-07-115 and 07-18-094 hybrids (Figure 2B). Conversely, the GSH content and the GSH/GSSG ratio were significantly lower in samples from the adult phase than that from the juvenile phase (Figure 2C,D). 

### 2.3. Changes in Chloroplast NADP^+^/NADPH Ratio

The ratio of NADP^+^/NADPH provides an indication of subcellular reducing power [45,46]. We calculated the NADP^+^/NADPH ratios to confirm or exclude the involvement of the NTRC/2-Cys Prx system during the vegetative phase change. The contents of NADP^+^, NADPH and NADP^+^+NADPH were significantly higher in samples from the adult phase than that from the juvenile phase, except in hybrid 07-18-094, where the differences in NADPH were not statistically significant (Figure 3A–C). The NADP^+^/NADPH ratio did not differ significantly between the adult and the juvenile samples (Figure 3D). These data indicated that the NTRC/2-Cys Prx system might not participate in plastid redox regulation.

### 2.4. Changes in ^1^O_2_ Production and ^1^O_2_ Related PNRS Gene Expression

The declines in expression of *MdMIR156a5*, *MdMIR156a12*, and miR156 along with node numbers were apparently shown in Figure 4A. It has been reported that the excess ^1^O_2_ release in PSII triggers an immediate induction of nuclear gene expression that depends on β-CC or on the plastid EX1 and EX2 proteins [28,47]. However, no significant difference in ^1^O_2_ levels were observed between leaves of the juvenile and the adult phase (Figure 4B). There are four genes in apple genome homologous to *Arabidopsis thaliana* and other Rosaceae plants encoding EX1/EX2 (Figure 4C). However, expression of none of these genes exhibited significant changes with node numbers (Figure 4D,E). 

Furthermore, we measured the expression of nuclear genes, *AAA-ATPase, GSTU13, LOX2, OPR3, SCL14, WRKY33, WRKY40, SIB1,* and *AOC4*. These genes are reported to be regulated at the transcriptional level by ^1^O_2_ triggered PNRS [25,27,28,29]. The expression of these homologs did not change with either miR156 or its precursors (Figure 4A,F). These data indicated that EX1/EX2 and the β-CC PNRS were independent of miR156 during the vegetative phase change.

### 2.5. Changes in Expression of PNRS Targeted Genes Generated by Photosynthetic Redox State, Secondary Metabolites, and Plastid Transcriptional Factor

The changes in plastid redox state can trigger PNRS, which are generated by GUN1, MEcPP, PAP, and WHIRLY [27,48]. We previously found that chloroplast H_2_O_2_ accumulated in the adult phase of apple [11]. To check whether these PNRSs were involved in miR156 regulation, the expression of some homologous genes targeted by these PNRS were analyzed. Many PhANGs have been shown to be targets of the GUN1 signaling pathway, including *ABI4*, *ACC2*, *GLK1*, *GLK2*, *LHCA1*, *LHCB1.1*, *LHCB1.2*, *SBPase,* and *cpFBPas*e [27,38,49]. Some nuclear genes, such as *APX2*, *BBX19*, *bZIP28*, *bZIP60*, *ZAT10*, are known to be regulated at the transcriptional level by MEcPP [27,33,50]. In addition, *ELIP2*, *APX2*, *RD29A* and *DREB2A* act downstream of the PAP signaling pathway [27,40,48]. Other genes, including *WRKY53*, *WRKY33*, *SAG12*, *NDHF*, *MER11*, *RAD50*, *POR*, and *NPR1*, are regulated by the WHIRLY signaling pathway [34,51]. The changes in expression of these apple homologous genes did not correlate with miR156 levels during the vegetative phase change (Figure 5). Therefore, none of these data implied the connection between miR156 expression and GUN1, MEcPP, PAP, and WHIRLY induced PNRS.

### 2.6. Changes in Tetrapyrroles and Related PNRS Gene Expression

In the apple genome, there are three *MdGUN5* genes encoding CHLH protein based on the amino acid alignment with CHLH of *Arabidopsis thaliana* and other Rosaceae plants like *Pyrus* (Figure 6D). *MdGUN5-1* (MD14G1072000) and *MdGUN5-2* (MD12G1073700) were predominantly expressed in apple leaves (Figure 6A). Both the FPKM values obtained and the relative expression of *MdGUN5-1* and *MdGUN5-2* increased gradually with node number (Figure 6B). Leaf Mg-Proto IX levels also increased gradually with node number in the three hybrids (Figure 7A). The correlation coefficient between Mg-Proto IX and miR156 relative expression was relatively high (r = –0.966 ~ –0.839, *p* < 0.05) (Figure 7A). 

Subsequently, we measured the expression of several nuclear genes including *GUN4*, *CBP*, *HY5*, *APRR5*, *ZTL*, *GLK1*, *GLK2*, *LHCB2.4*, *LHCB1.1,* and *RBCS.* Because the expression of these genes was regulated by Mg-Proto IX signaling [38,39,40]. Of the apple homologues of these genes, *RBCS, LHCB2.4, ZTL,* and *HY5* were upregulated along with node numbers, which were consistent with the changes in Mg-proto IX content. The expression of these genes was slightly upregulated along with the node numbers (Figure 6F), which was negative with the changes in miR156 expression levels. These data together indicated an intimate association between Mg-Proto IX biosynthesis and miR156 expression.

Two *GUN6* homologs, *MdGUN6-1* (MD08G1044400) and *MdGUN6-2* (MD15G1061700), were identified in the apple genome (Figure 6E). *MdGUN6-1* was stably expressed in leaves, while *MdGUN6-2* showed a progressive decrease in expression along with node numbers (Figure 6A,C). In 07-07-115, leaf heme contents declined between 1–150 nodes. No obvious changes in heme content were detected in the other two hybrids. The correlation coefficients between heme contents and miR156 expression were as low as r = 0.234 ~0.561 (*p* > 0.05) (Figure 7B). Hence, the possible association between heme and miR156 was negated.

### 2.7. Validation of the Relationship between Tetrapyrroles and miR156 Expression or GSH Content

To validate the correlation between tetrapyrroles levels, miR156 expression, or GSH content, ALA and gabaculine were applied to the culture media of in vitro apple shoots. ALA is a precursor of tetrapyrroles and is readily taken up by plants. Supplying ALA induces rapid increase in tetrapyrrole biosynthesis [41]. Gabaculine inhibits the activity of the second ALA synthesis enzyme, glutamate semialdehyde aminomutase, leading to decrease in tetrapyrroles [52]. 

Mg-Proto IX levels increased significantly in in vitro shoots grown on ALA containing medium by 2-3 days after treatment (Figure 8A). No robust significant changes in GSH content and relative miR156 expression were detected in response to ALA or gabaculine treatments compared to untreated control (Figure 8A,B). On the fourth day after ALA treatment, GSSG content in treated shoots was decreased significantly compared to the control. On the second day after ALA treatment, the GSH+GSSG content was higher (Figure 8A). On the 6th and 10th days after gabaculine treatment, the GSSG content in treated shoots was significantly higher than that in the control (Figure 8B). The relative expression levels of *MdHY5*, *MdLHCB2.4* and *MdZTL*, which are target nuclear genes of Mg-Proto IX, increased after ALA treatments and decreased after gabaculine treatments compared with the untreated control (Figure 8A,B). 

To investigate whether the miR156 regulated tetrapyrrole-PNRS, Mg-Proto IX and heme contents were measured in apple transgenic lines of OEMdMIR156a6 and 35S::MIM156. Mg-Proto IX and heme concentrations decreased in OEMdMIR156a6 and increased significantly in 35S::MIM156 (Figure 9A,B). The expression of *MdGUN5* and *MdGUN6* genes was substantially higher in the 35S::MIM156 transgenic plants compared with the wild type (WT). However no significant changes in *MdGUN5* and *MdGUN6* genes occurred in the OEMdMIR156a6 transformants, except for significant down regulation of *MdGUN5* and *MdGUN6*-2 genes in line 3 (Figure 9C). In OEMdMIR156a6 and 35S::MIM156 transformants, the changes in the expression of tetrapyrrole-PNRS target genes were consistent with the changes in Mg-Proto IX concentrations (Figure 9D).

Perturbation of cellular thiol-redox homeostasis has a major effect on plant development, whereas GSH is a general redox signaling transducer [15]. We examined whether tetrapyrrole-PNRS was affected by the GSH status by using GL-3 transgenic lines over-expressing gamma-glutamyl transferase gene1 (OEMdGGT1) and MdGGT1-RNAi lines. It is reported that GSH levels and miR156 expression are elevated in OEMdGGT1 and decreased in MdGGT1-RNAi lines [22]. We found that Mg-Proto IX and heme levels were relatively lower in OEMdGGT1 lines but were significantly higher in MdGGT1-RNAi lines compared with WT (Figure 9A,B). The expression of *MdGUN5* and *MdGUN6* genes was highly reduced only in line OEMdGGT1-1, and the expression of *MdGUN5-1* and *MdGUN5-2* was remarkably increased in MdGGT1-RNAi lines. (Figure 9C). The expression of several tetrapyrrole-PNRS regulated genes was also slightly inhibited or promoted in the OEMdGGT1 and MdGGT1-RNAi transformants, respectively (Figure 9D).

## 3. Discussion

### 3.1. The Accumulation of Chloroplastic H_2_O_2_ in the Adult Phase is Due to Decreased Scavenging Capacity rather than Higher Photosynthetic Activity

Whenever light exceeds the photosynthetic capacity, a light-induced photoinhibition occurs in PSII and results in electron leakage to form superoxide anion radical, which dismutes to H_2_O_2_ [53]. Higher photosynthetic rate causes higher ROS release in chloroplasts, whereas strong ROS generating or attenuated ROS scavenging leads to inevitable modifications of PNRS [19,20]. A large class of photosynthetic genes was found to be induced in the juvenile phase of maize (*Zea mays* L.), priming energy production for vegetative growth [54]. Similarly, in apple seedlings, several photosynthesis-associated proteins or genes encoding photosynthesis-related proteins are also expressed abundantly in the juvenile phase [21,55]. However, the photosynthetic rate of transgenic tobacco (*Nicotiana tabacum* L.) mimicry156 plants was significantly higher than the control due to more chlorophyll. Conversely, miR156 over-expressing plants had much lower rates of photosynthesis and less chlorophyll, indicating an involvement of miR156 in chlorophyll biogenesis or degradation [56]. In English ivy (*Hedera helix*), the higher photosynthetic capacity in adult than juvenile cuttings was due to thicker leaves with more chloroplasts/unit leaf area, rather than enhanced photosynthetic rate [57]. By comparing scions from the juvenile and the adult phases of the same hybrid, we observed that the photosynthetic properties did not vary significantly between ontogenetic phases (Figure 1). Therefore the adult-phase-specific plastid H_2_O_2_ accumulation may be attributed to shift in scavengers or other factors, excluding changes in photosynthetic rate, because developmental cues like vegetative phase change or physiological aging accelerates ROS production [58].

The NADP^+^/NADPH ratio is a marker for ROS quenching ability by the plastid NTRC/2-Cys Prx system [24]. We found that the NADP^+^/NADPH ratio did not vary significantly with node numbers. This indicated that the NTRC/2-Cys Prx level was robustly maintained during the phase change (Figure 3). In this study, both GSH content and the GSH/GSSG ratio decreased significantly during ontogenesis, which supported our previous hypothesis that a shift in major antioxidants from GSH to ASC during the phase change attenuated the cellular ROS scavenging ability [11,18,22]. This was consistent with that GSH plays a key role in ASC–GSH coupled non-enzyme ROS scavenging in plant cells [59].

### 3.2. The PNRS Acts Concomitantly or Downstream of miR156 During Ontogenesis

In stressful environments, plastids act as environmental sensors and communicate with the nucleus via NPRS including EX1/EX2, β-CC, MEcPP, PAP, tetrapyrrole etc. [27,40]. ^1^O_2_ triggers EX1/EX2 and β-CC production to regulate the expression of nuclear genes [27]. No substantial differences were detected between the juvenile and the adult phase in ^1^O_2_ levels, expression of EX1/EX2 genes, and their PNRS genes. Similarly, most of the target genes of PAP, MEcPP, GUN1, and WHIRLY did not change with node numbers and miR156. These data excluded the involvement of these NPRS in miR156 regulation. We did not analyze the expression of *GUN1* and *WHIRLY* genes, because they are known to be post-transcriptionally regulated in response to stress [31,51].

Tetrapyrrole-PNRS, such as Mg-Proto IX, interferes with the stability of ZTL, and ZTL negatively regulates the transcription factor HY5, which in turn regulates PhANGs [39,60]. Mg-Proto IX content, the expression of its synthesis gene *MdGUN5*, and the expression of Mg-Proto IX PNRS genes increased gradually with node numbers. Mg-Proto IX content was negatively correlated with miR156 level. However, altering Mg-Proto IX and heme contents via exogenous ALA or gabaculine applications caused consistent changes in Mg-Proto IX content and the expressions of the nuclear genes, but did not affect miR156 expression. Manipulation of miR156 levels using *MdMIR156a6*, *MdGGT1,* and *MIM156* transgenic lines led to corresponding changes in not only the expression of tetrapyrrole synthesis genes, tetrapyrrole content, but also their target nuclear genes. Therefore, the Mg-Proto IX PNRS was regulated downstream of miR156, irrespective of adult-phase-specific plastid H_2_O_2_ accumulation.

### 3.3. GSH Acts as a General Master Signaling Molecule to Fine Tune Redox Homeostasis and Subsequently the miR156 Expression

GSH is a dual function molecule, on one hand precisely fine tunes cellular redox homeostasis, and on the other hand acts as a general redox signaling transducer [61,62]. GSH participates in the transmission of redox signals to the nucleus and activates jasmonic and salicylic acid accumulation [34,63,64]. Abiotic stress-induced MEcPP accumulation may be converted to general redox signals by changing the GSH/GSSG ratio [40,65]. β-CC can strongly induce the conjugation of GSH with reactive electrophile species, which induces nuclear gene expressions [26].

Using grafted young trees of apple hybrids, we observed that the GSH content and GSH/GSSG ratio were significantly lower in adult phase than that in juvenile phase. Manipulation of cellular GSH content or the GSH/GSSG ratio via GSH precursor or inhibitor, as well as via transgenic plants, led to corresponding changes in the expressions of *MdMIR156a5*, *MdMIR156a12* and mature miR156 [11,22].

The mechanism underlying the promotion of miR156 expression by GSH is not fully understood, but recent studies have suggested that protein S-glutathionylation affects the expression of downstream genes. For example, GSH has been shown to increase the binding of WRKY33 to the promoter of target genes such as 1-aminocyclopropane-1-carboxylate synthase [66]. The direct interaction between GSH and miR156 precursors should be one objective of the future study.

## 4. Materials and Methods

### 4.1. Plant Materials

Three ten-year-old hybrid trees with their own roots, raised from hybrid seeds of a cross *Malus asiatica* ‘Zisai Pearl’ × *M. domestica* ‘Red Fuji’, were used in this study. Given the hyper-heterozygosity of the parents and extensive phenotype segregation among the hybrids, three intact hybrid trees, 07-07-115, 07-07-119 and 07-18-094, were sampled as three biological replicates [11,22]. Young leaves were sampled from 1-year-old suckers and annual branches. The samples taken from the first to the 80th nodes were defined as the juvenile phase, and those from the 120th node to the canopy top as the adult phase [4,11,18]. By using these three hybrid trees, the expression of related genes and the tetrapyrrole contents were determined. The hybrid 07-07-115 was used for a transcriptome analysis.

To confirm the robustness of photosynthetic activity in both juvenile and adult leaves, branches from the canopy top and suckers from the basal part of the trunk were used as scions and grafted onto M26 rootstocks (60 plants each), representing the adult and juvenile phase, respectively. The scions grafted on M26 rootstock were used to measure the photosynthetic rate, chlorophyll fluorescence, contents of ROS, glutathione and NADPH. 

Four transgenic materials were used for experiments; they are 35S::MIM156, OEMdMIR156a6, OEMdGGT1, and MdGGT1-RNAi. An expression construct contains the miR156 precursor MdMIR156a6 (MDC018927.245) and the *Cauliflower mosaic virus* 35S promoter (OEMdMIR156a6). An artificial target mimics were generated by modifying the sequence of the AtIPS1 gene to inhibit the activity of miR156 (35S::MIM156). Transformation of apple was conducted by *Agrobacterium*-mediated transformation system using in vitro cultured leaflets of GL-3 as explants [67]. The OEMdGGT1 and MdGGT1-RNAi transgenic GL-3 lines were generated previously [22].

### 4.2. Chemical Treatments and Sampling

To manipulate Mg-Proto IX and heme levels, in vitro shoots of 07-07-115, grown on Murashige and Skoog (MS) media with 0.5mg/l IBA and 0.3 mg/l 6-BA, were treated with 5-aminolevulinic acid (ALA) (0 mM, 0.05 mM, 0.5 mM and 5 mM) or gabaculine (0 μM, 20 μM, 40 μM and 80 μM). These chemicals were added to the sterilized medium by autoclaving [52]. Fifteen bottles of in vitro shoots (5 plants per bottle) were treated for each chemical concentration. The samples were collected 0 h, 12 h, 24 h, 36 h and 48 h after treatment with ALA and 0 d, 1 d, 3 d, 5 d, and 7 d after treatment with gabaculine.

### 4.3. Photosynthetic Rate and Chlorophyll Fluorescence Assay

From 10:00 am to 11:00 am on sunny days, June 2018, the net photosynthetic rate (NPR) was analyzed using the fourth to sixth fully expanded leaves of the juvenile and the adult scions grafted onto M26 rootstocks. Fifteen leaves were measured for either juvenile phase or the adult phase scions per hybrid. The analysis was performed using a LI6400 photosynthesis system (LI-COR, Lincoln, USA). Chlorophyll fluorescence was measured using an imaging pulse amplitude modulated (PAM) fluorometer (IMAG-MAX/L; Walz, Germany).

### 4.4. Quantification of H_2_O_2_

H_2_O_2_ was quantified using the method described by Du et al. [18]. A leaf sample of 200 mg was ground to a fine powder in liquid nitrogen. H_2_O_2_ was extracted thoroughly using 1.9 mL pre-cooled acetone, then centrifuged (3000× *g*) at 4 °C for 20 min. Then, 5% titanium sulfate and ammonia were added to the supernatant, and it was centrifuged again (3000× *g*) at 4 °C for 10 min. The precipitant was rinsed three to five times with acetone and vortexed. It was then redissolved in 1.0 M H_2_SO_4_. The absorbance of the supernatant at 415 nm was measured using an UV spectrophotometer (UV 1800, Shimadzu, Japan) against a blank. The H_2_O_2_ content was determined based on a standard curve plotted using known H_2_O_2_ concentrations. 

### 4.5. Quantification of GSH and GSSG Content and the GSH/GSSG Ratio

Glutathione concentration was quantified using a GSH and GSSG assay kit (Beyotime, Nantong, China) following the manufacturer’s instructions. GSH and GSSG measurements were performed using a microplate reader (Model 680, Bio-Rad, Hercules, CA, USA).

### 4.6. Quantification of ^1^O_2_

Oxygen singlet oxygen concentration of leaves was quantified using an Oxygen singlet oxygen colorimetric quantitative detection kit (SCIENTIFICS INC. U.S.A). Oxygen singlet oxygen measurements were performed using a microplate reader (Model 680, Bio-Rad, Hercules, CA, USA).

### 4.7. Quantification of NADP^+^ and NADPH Contents and Their Ratio

NADPH was quantified using a NADP^+^/NADPH Assay Kit with WST-8 (Beyotime, Nantong, China). The measurements of NADP^+^ and NADPH levels were performed using a microplate reader (Model 680, Bio-Rad, Hercules, CA, USA).

### 4.8. Quantification of the Mg-Proto IX Concentration

Mg-Proto IX was analyzed by high performance liquid chromatography method according to Strand et al. [37]. Leaf material (0.2 g) was homogenized in 1 mL acetone (0.1 M NH_4_OH, 9:1 [*v*/*v*]) and homogenized and centrifuged with ice-cooling. The residue was resuspended again, and the same procedures described above were repeated. The collected supernatants were mixed and centrifuged prior to high performance liquid chromatography (Waters, Milford, USA). The eluate (0–10 min) was passed through an ACQUITY Ultra Performance Liquid Chromatography (UPLC) Fluorescence (FLR) Detector (Waters, Milford, USA), with initial excitation wavelength of 417 nm and an emission wavelength 595 nm, and secondary excitation and emission at 402 nm and 633 nm, respectively [68]. Mg-Proto IX was identified and quantified using authentic standards.

### 4.9. Determination of Heme Concentration

Samples (0.1 g) from fresh leaves were homogenized with a pestle and a mortar in 1mL of Phosphate Buffer Saline (PBS), pH 7.4, centrifuged at 3000× *g*, at 4 °C for 20 min. The supernatant was used for the assay. Heme concentration was quantified using a Plant Fe-protoporphyrin (heme) ELISA Kit (Jonln, Shanghai, China). Heme measurements were performed using a microplate reader (Model 680, Bio-Rad, Hercules, CA, USA).

### 4.10. Analysis of the Relative Expression of miR156 and Gene Family Members

Total RNA and microRNA were extracted from leaf samples of hybrids (07-07-115, 07-07-119, and 07-18-094) with their own roots. Total RNA was extracted using a modified cetyltrimethylammonium bromide method. Complementary DNA was synthesized using a cDNA Synthesis Kit (Takara). MicroRNA was extracted by using the RNAiso for Small RNA kit (9753Q, Takara, Dalian, China) according to the manufacturer’s instruction. To analyze the relative gene expression, the quantitative RT-PCR was performed for mature miR156, the precursor genes, and some protein coding genes using SYBR green reagents (RR820A, Takara, Dalian, China) in an Applied Biosystems 7500 real-time PCR system [22]. The primer pairs used are listed in Appendix A. The DNA and mRNA sequences of all genes were obtained from the apple genome website (http://genomics.research.iasma.it/) using a basic local alignment search tool (BLAST) search. β-Actin (MD07G1012400) was used as the reference gene.

A transcriptome analysis of young leaves from hybrid 07-07-115 was performed using Illumina next generation sequencing. Total RNA was extracted from young leaf samples of suckers per 30 internodes (1~180 internodes) along the central leading cane, in three biological replicates. RNA-seq library construction, sequencing, and bioinformatics analysis were performed as described in Huang and colleagues [69]. Sequencing was performed using a paired-ends 150 strategy on an Illumina HiSeq 2500 system. The complete data set and main results will be published elsewhere soon (BioProject ID: PRJNA580040). The expression (in fragments per kilobase of transcript per million mapped reads (FPKM)) of genes included in this study, based on the UniGene assembly, is shown in Appendix A.

### 4.11. Statistical Analysis

All the experiments were designed with three biological replicates and three technical replicates, except that no technical replicates were used in the transcriptomic analysis, and the assays of photosynthetic rate and chlorophyll fluorescence were designed with fifteen biological replicates and three technical replicates. Statistical analysis was performed using the Statistical Product and Service Solutions (SPSS) software (IBM Co., Armonk, NY, USA). All experimental data were tested by a Student’s *t*-test or a Duncan’s multiple-range test.

### 4.12. Phylogenetic Analysis

*Arabidopsis thaliana* protein sequences were downloaded from the *Arabidopsis thaliana* website (http://www.arabidopsis.org/). Protein sequences from other species were obtained using a Basic Local Alignment Search Tool Protein (BLASTP) search of known *Arabidopsis thaliana* protein sequences against the Apple Genome Database (https://www.rosaceae.org/). Protein sequences were compared by MEGA7 were used to construct phylogenetic graphs [70]. The algorithm used for the tree generation was neighbor-joining (NJ) with 1000 bootstrap repeats.

## 5. Conclusions

The adult-phase-specific plastid H_2_O_2_ accumulation may attribute to the shift in scavengers or other factors excluding changes in photosynthetic rate. Singlet oxygen, GUN1, MEcPP, PAP, and WHILY1 generated PNRS were not involved in ontogenesis of apple seedlings. The Mg-Proto IX PNRS was regulated downstream of miR156, irrespective of adult-phase-specific plastid H_2_O_2_ accumulation.

## Figures and Tables

**Figure 1 plants-09-00653-f001:**
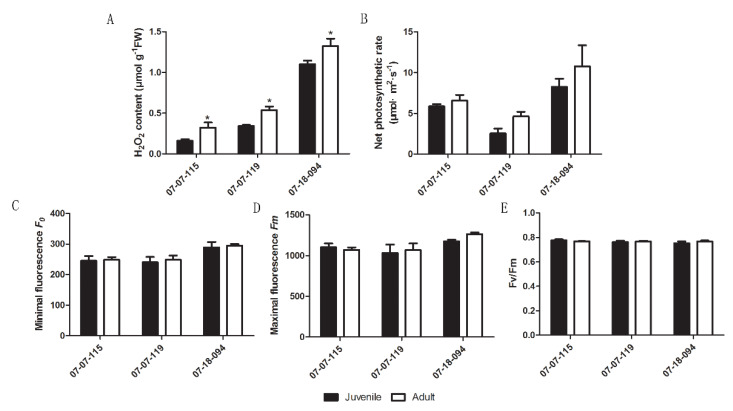
Comparison of leaf H_2_O_2_ content (**A**), net photosynthetic rate (**B**) and photosynthetic electron transport properties (**C**–**E**) in scions from the juvenile and the adult phase hybrids derived from *Malus asiatica* ‘Zisai Pearl’× *M. domestica* ‘Red Fuji’ grafted onto M26 rootstocks. Numbers beneath the horizontal axes indicate the three hybrids. Error bars represent the standard deviations of three biological replicates. Asterisks indicate statistical significance at *p* < 0.05.

**Figure 2 plants-09-00653-f002:**
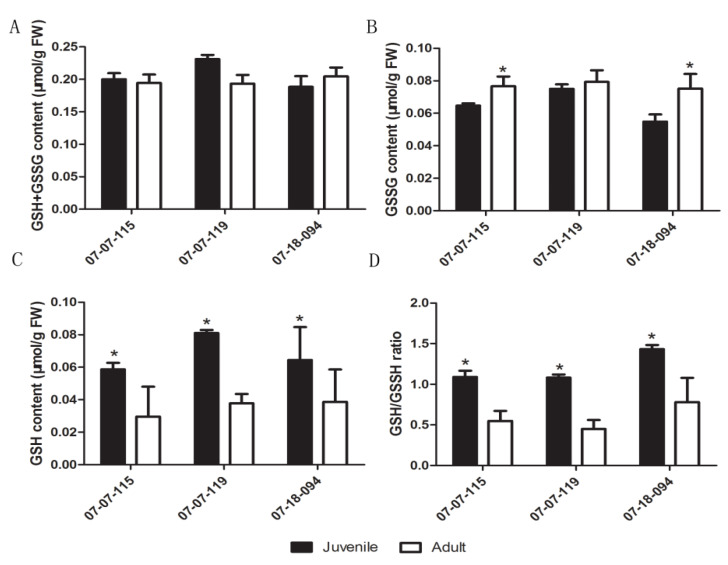
Comparison of GSH+GSSG (**A**), glutathione disulfide (GSSG) (**B**), glutathione (GSH) contents (**C**) and the GSH/GSSG ratio (**D**) between the juvenile and the adult phase scion leaves from hybrids derived from Malus asiatica ‘Zisai Pearl’× M. domestica ‘Red Fuji’ grafted onto M26 rootstocks. Numbers beneath the horizontal axes indicate the three hybrids. Error bars represent the standard deviation of three biological replicates. Asterisks indicate statistical significance at *p* < 0.05.

**Figure 3 plants-09-00653-f003:**
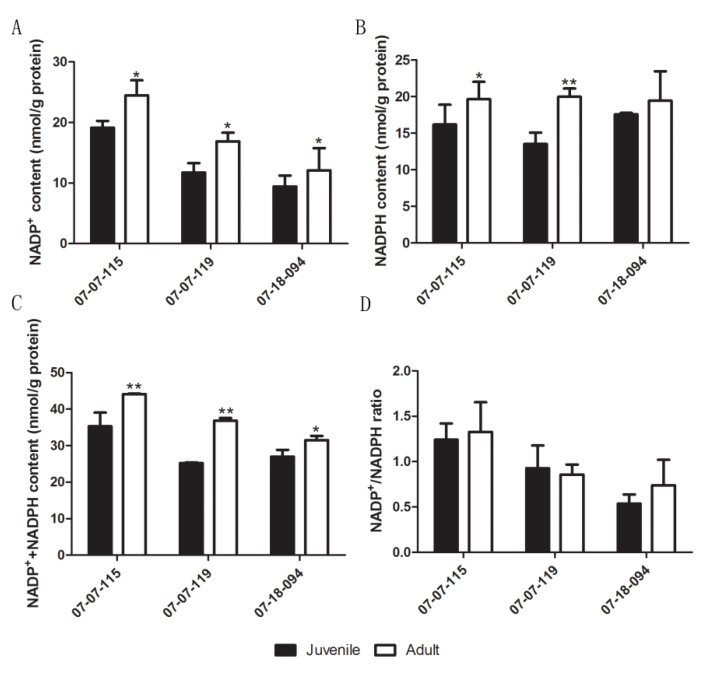
Comparison of NADP^+^(**A**), reduced and oxidized nicotinamide adenine dinucleotide phosphate (NADPH) (**B**), NADPH + NADP^+^ (**C**) contents and the NADP^+^/NADPH ratio (**D**) between leaves of the juvenile and the adult phase scions of hybrids derived from *Malus asiatica* ‘Zisai Pearl’× *M. domestica* ‘Red Fuji’. Numbers beneath the horizontal axes indicate the three hybrids. Error bars represent the standard deviation of three biological replicates. One asterisk indicates statistical significance at *p* < 0.05, two asterisks indicate statistical significance at *p* < 0.01.

**Figure 4 plants-09-00653-f004:**
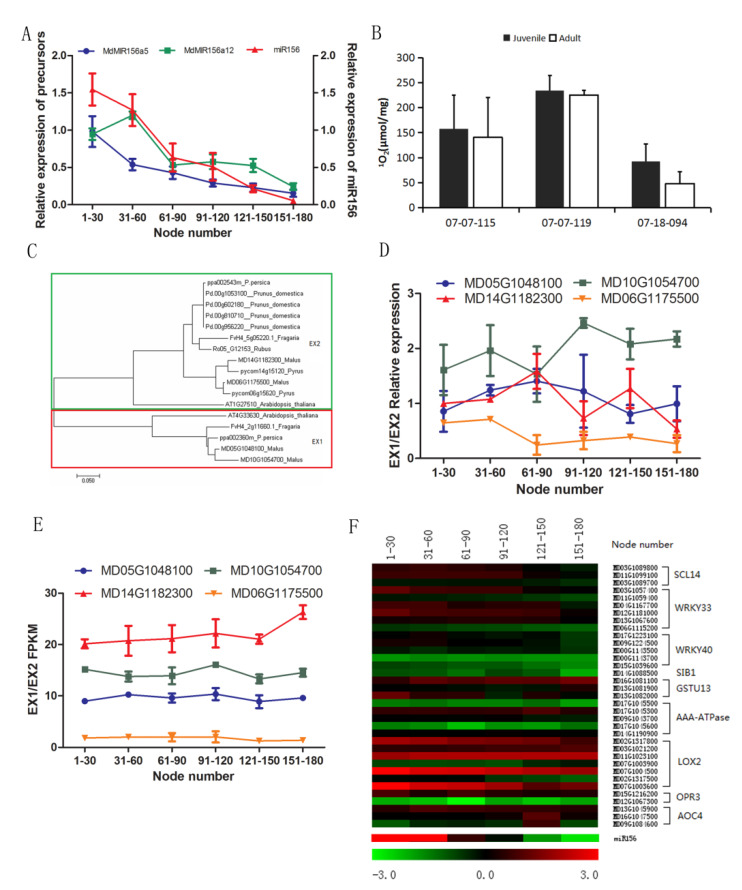
Changes in singlet oxygen (^1^O_2_) production, expressions of ^1^O_2_-related plastid-nucleus retrograde signaling genes, microRNA156 (miR156) and the precursors during ontogenesis in leaves of apple hybrids derived from *Malus asiatica* ‘Zisai Pearl’× *M. domestica* ‘Red Fuji’. (**A**) Dynamic changes in expression of miR156 and its precursor genes alongside internode numbers of the hybrid 07-07-115. (**B**) Comparison of ^1^O_2_ content in the juvenile and the adult phase leaves of three hybrids (07-07-115, 07-07-119, and 07-18-200). (**C**) Phylogenetic relationship of EX1/EX2 genes from apple-related species. (**D**) Changes in relative EX1/EX2 expression with node number in a hybrid (07-07-115) assayed by quantitative real time (qRT)-PCR. (**E**) Changes in fragments per kilobase of transcript per million fragments mapped (FPKM) values indicating expression of EX1/EX2 gene family members. (**F**) Heat map showing the expression of nuclear genes downstream of EX1/EX2 and the β-CC signaling pathway. The dynamic changes in miR156 expression along with node numbers were presented alone at the bottom of the heat map to make convenience for comparison. Error bars in (**A**,**B**,**D**,**E**) represent the standard deviation of three biological replicates.

**Figure 5 plants-09-00653-f005:**
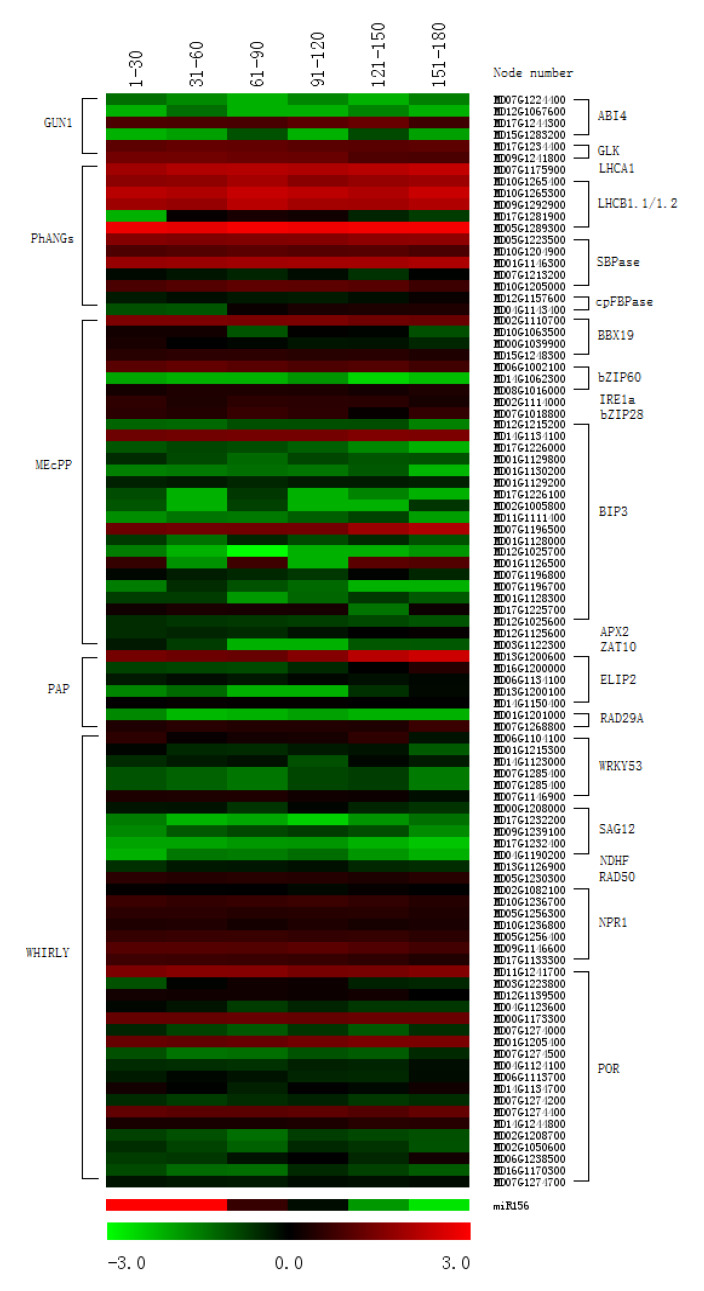
Heat map showing photosynthetic redox state-generated plastid to nucleus retrograde signal targeted gene expression profiles during ontogenesis in leaves of apple hybrids derived from *Malus asiatica* ‘Zisai Pearl’× *M. domestica* ‘Red Fuji’. The gene expression was shown in fragments per kilobase of transcript per million fragments mapped (FPKM) of nuclear genes downstream from GUN1, PhANGs, MEcPP, PAP and the WHIRLY signaling pathways. The dynamic changes in miR156 expression, along with node numbers, were presented alone at the bottom of the heat map to make convenience for comparison.

**Figure 6 plants-09-00653-f006:**
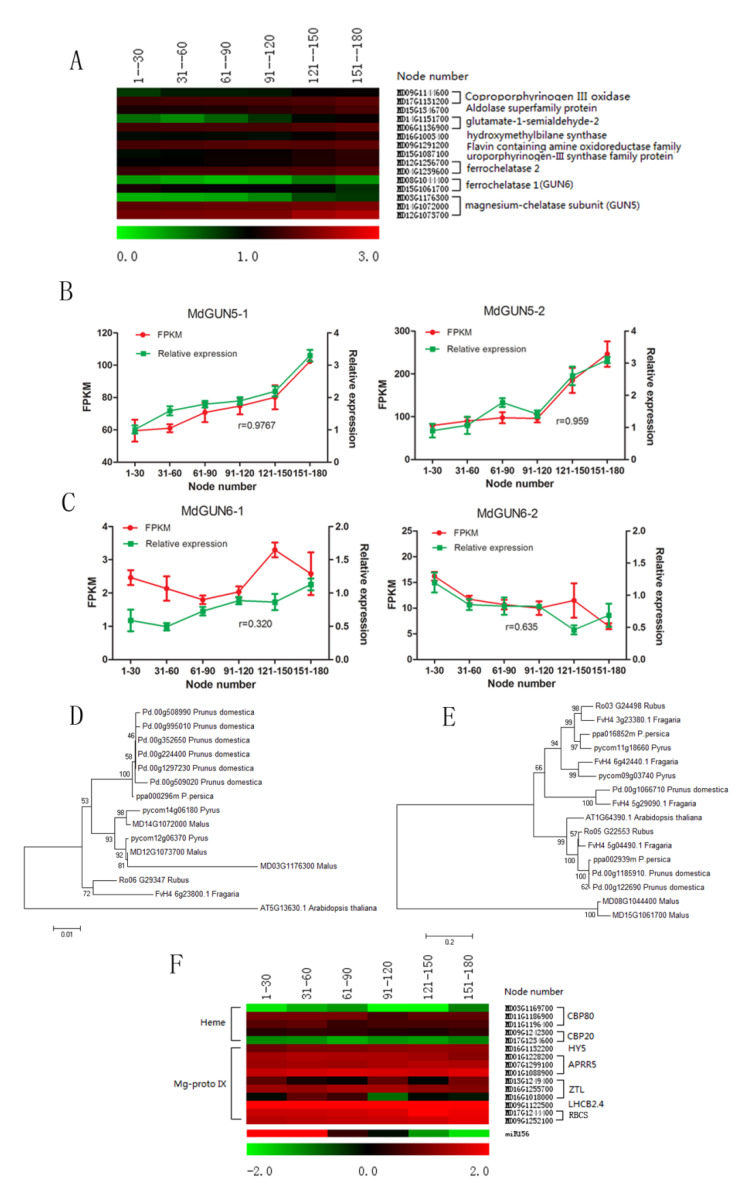
Mg-Proto IX and heme biosynthetic genes and their target nuclear genes during ontogenesis of apple hybrid 07-07-115 derived from *Malus asiatica* ‘Zisai Pearl’× *M. domestica* ‘Red Fuji’. (**A**) Heat map showing expression of Mg-Proto IX and heme biosynthetic genes based on RNA sequencing (RNA-seq) data. (**B**,**C**) Validation of *MdGUN5* (**B**) and *MdGUN6* (**C**) gene expression using RNA-seq in fragments per kilobase of transcript per million fragments mapped (FPKM) and relative expression by quantitative real time q(RT)-PCR. Error bars represent one standard deviation of the three biological replicates. The r-value represents the Pearson correlation coefficient between FPKM and the relative expression. (**D**,**E**) Phylogeny of *MdGUN5* and *MdGUN6* gene family members, respectively. The phylogenetic relationship was inferred by neighbor joining analysis based on amino acid differences (p-distance) using the MEGA 7.0 program with complete deletion and 1000 bootstrap replicates. The short line at the bottom of the panel represents the scale bar. (**F**) Expression profiles of Mg-Proto IX and heme target nuclear genes based on RNA-seq analysis.

**Figure 7 plants-09-00653-f007:**
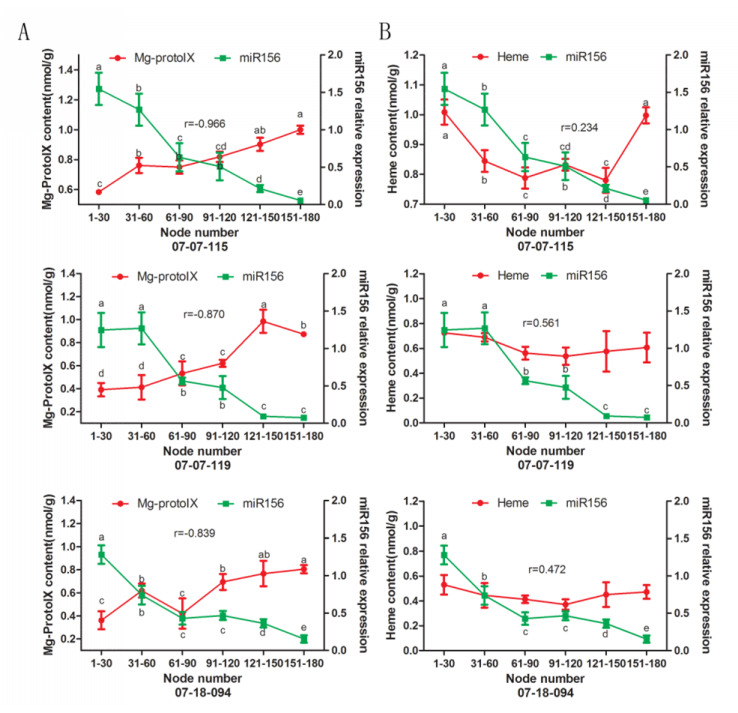
Changes in leaf Mg-Proto IX (**A**) and heme (**B**) contents and the correlation with miR156 levels, along with node numbers of hybrids derived from *Malus asiatica* ‘Zisai Pearl’× *M. domestica* ‘Red Fuji’. The r-value represents the Pearson correlation coefficient between miR156 relative expression and Mg-Proto IX or heme contents. Error bars represent one standard deviation of the three biological replicates. The lower-case letters above each column indicate the statistical significance (*p* < 0.05) by analysis of variance followed by Duncan’s multiple-range test.

**Figure 8 plants-09-00653-f008:**
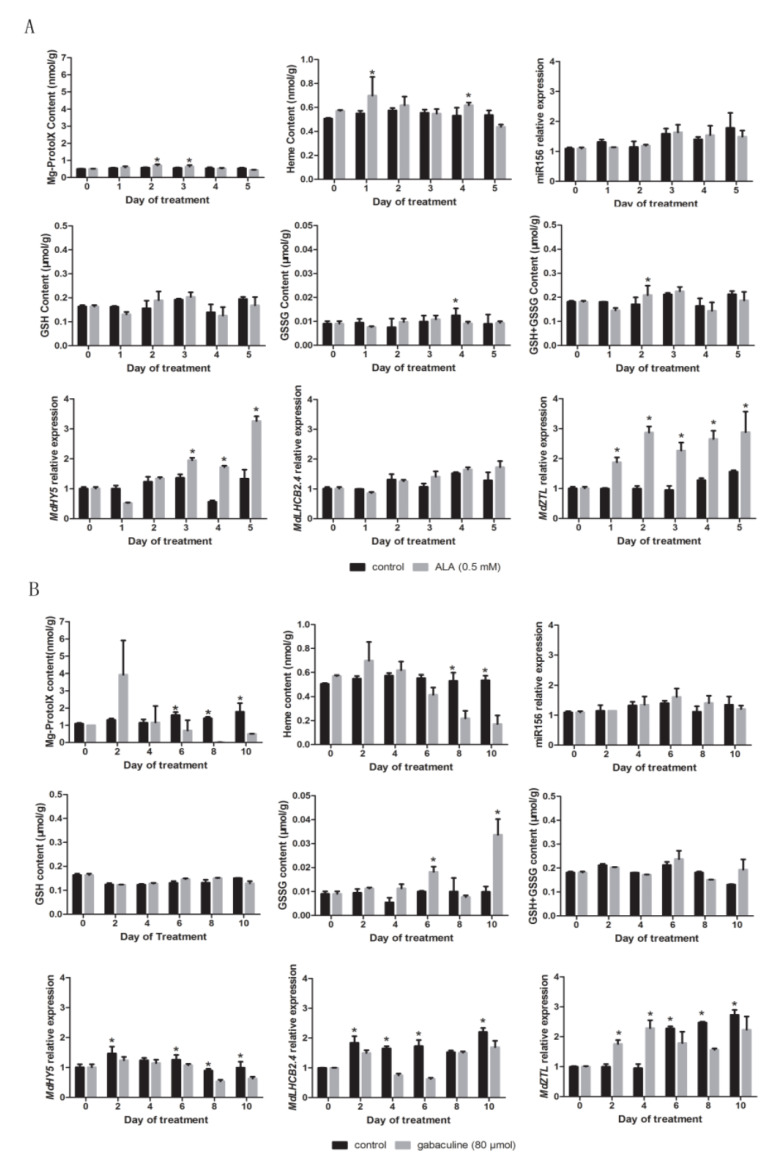
Changes in Mg-Proto IX and heme concentration, glutathione (GSH), glutathione disulfide (GSSG) contents, and relative expression of miR156 and Mg-Proto IX related nuclear genes (*MdHY5*, *MdLHCB2.4*, and *MdZTL*) in in vitro shoots of apple hybrids (*Malus asiatica* ‘Zisai Pearl’ × *M. domestica* ‘Red Fuji’) treated with 5-aminolevulinic acid (ALA) (**A**) and gabaculine (**B**). Error bars represent the standard deviations of three biological replicates. Asterisks indicate statistical significance at *p* < 0.05.

**Figure 9 plants-09-00653-f009:**
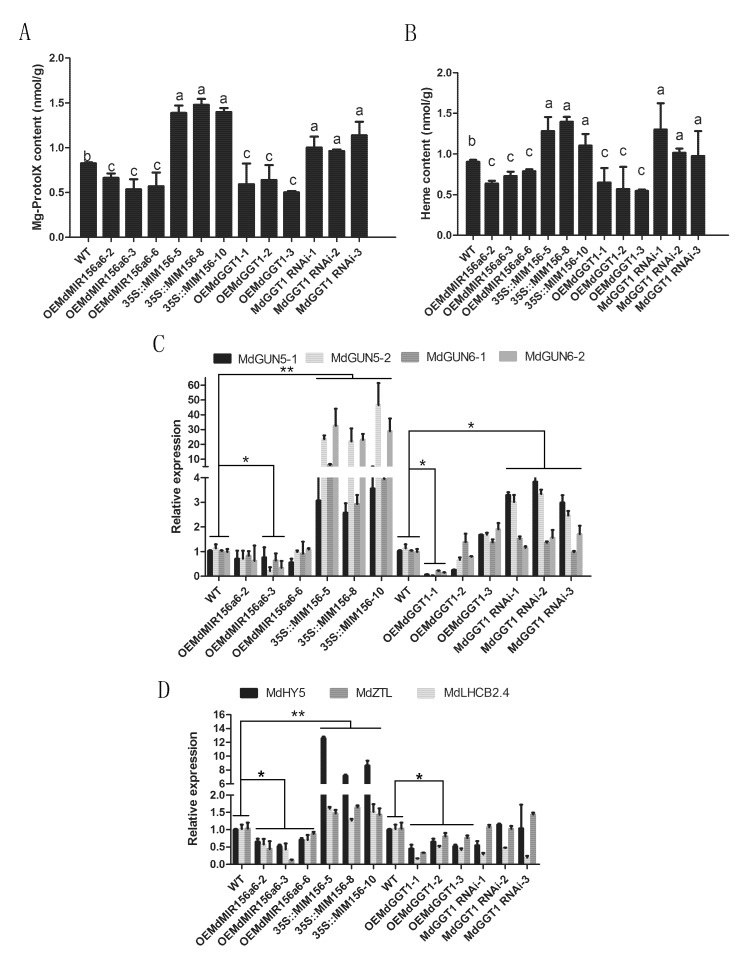
Changes in Mg-proto IX (**A**) and heme (**B**) concentrations and their synthesis (**C**) and target (**D**) gene expression in transformed GL-3 in vitro shoots constitutively over-expressing MIM156, *MdMIR156a6*, *MdGGT1* or MdGGT1-RNAi compared with the untransformed wild type (WT). Error bars represent the standard deviations of three biological replicates. Different lowercase letters and asterisks indicate statistical significance at *p* < 0.05, two asterisks indicate statistical significance *p* < 0.01.

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
