# Peer review of "MicroRNA156 (miR156) Negatively Impacts Mg-Protoporphyrin IX (Mg-Proto IX) Biosynthesis and Its Plastid-Nucleus Retrograde Signaling in Apple"

_plants, 2020, doi:10.3390/plants9050653_

Round 1
Reviewer 1 Report
The authors evaluate several key players in the juvenile-adult transition of apple, showing non-involvement of some of them and the role of miR156 in the biosynthesis of Mg-proto IX and its PNRS. It is an interesting study with valuable information that continues their previous work (Jia et al 2017, Du et al 2015). The experiments have been performed rigorously with enough replications and sample sizes and support the conclusions. I consider this manuscript suitable for Plants but minor corrections need to be done before its publication
Line 60. Remove the repeated `in´.
Line 230. Capital letter H in `however´.
Line 267. While the correlation between Mg-ProtoIX and mR165 are clearly observed in Fig 7A, the genes RBCS, LHCB2.5, ZTL, and HY5 are only upregulated along with the node numbers (Fig 6F) without a clear correlation with miR165 levels. miR165 shows up and downregulation along with the node numbers (Fig 6F). Thus, the sentence of line 267 should be in accordance with the results.
Line 279. Figure 6. For clarity, the authors should add that the phylogenetic relationships among MdGUN5 and MdGUN6 gene family members were inferred by neighbor-joining analysis. The authors should indicate which represents the scale bar.
Line 389. `.. and etc?. `and´ should be removed.
Author Response
Dear Reviewer,
We thank you for your thoughtful suggestions and insights, which have enriched the manuscript and produced a more balanced and better account of the research.
According to your suggestions, we have made modifications in the manuscript . The details are as follows:
Comment 1: Line 60. Remove the repeated `in´.
Authors’ response: Revised accordingly.
Comment 2: Line 230. Capital letter H in `however´.
Authors’ response: Revised accordingly. That was actually at line 203 of the former version.
Comment 3: Line 267. While the correlation between Mg-ProtoIX and mR165 are clearly observed in Fig 7A, the genes RBCS, LHCB2.5, ZTL, and HY5 are only upregulated along with the node numbers (Fig 6F) without a clear correlation with miR165 levels. miR165 shows up and downregulation along with the node numbers (Fig 6F). Thus, the sentence of line 267 should be in accordance with the results.
Authors’ response: We think this comment is quite helpful for us to improve the accuracy of presentation in the manuscript. We rephrased the statement to make accuracy. “The expression of these genes was slightly upregulated along with the node numbers (Figure 6F), which was negative with changes in miR156 expression levels.”
Comment 4: Line 279. Figure 6. For clarity, the authors should add that the phylogenetic relationships among MdGUN5 and MdGUN6 gene family members were inferred by neighbor-joining analysis. The authors should indicate which represents the scale bar.
Authors’ response: We appreciated this comment and made revision accordingly to make the presentation better. Following the reviewer’s comment, we added the construction method of phylogenetic tree and scale bar annotation in Figure 6 legend.
Comment 5: Line 389. `.. and etc?. `and´ should be removed.
Authors’ response: Revised accordingly.
We’re looking forward to your reply.
Sincerely,
Qingbo Zheng
Reviewer 2 Report
The study by Zheng and co-workers is aimed at elucidating more in depth the molecular triggers activating the plastid-nucleus retrograde signaling (PNRS) system and its related influence on the plant growth and development. Based on the notion that reactive oxygen species can stimulate the PNRS machinery, and on the fact that low expression of the miR156 in adult apple leaves is coupled with increased hydrogen peroxide accumulation in the chloroplast, the authors tested whether either adult-phase-specific increase in chloroplast H2O2-induced PNRS is able to influence miR156 expression, or alternatively whether miR156 can stimulate plastid ROS accumulation and subsequent PNRS during the ontogenesis process.
Hence, in search for a functional link between miR156 and establishment of PNRS, expression changes of mature miR156 and its precursors and of PNRS component genes were investigated using RNAseq and target RTqPCR approaches, making comparison of adult and juvenile-phase leaves taken from three Malus hybrids. These data were further integrated with measurements of assimilation rates and photosynthesis efficiency in the same samples accompanied by quantification of ROS species (H2O2 and O2), ROS scavenging effectors (GSH, GSSG, NADP+, NADPH, and corresponding ratios) and of known PNRS-associated molecules, such as Mg protoporphyrin IX (Mg-Proto IX) and Fe-protoporphyrin IX (heme). Additionally, Mg-Proto IX and heme content was analyzed in transgenic apple lines overexpressing or silencing miR156 and put in correlation with expression profiles of their biosynthetic or target genes. The relationship between tetrapyrroles and miR156 expression or GSH content was also discussed in light of results of chemical treatments addressed to alter Mg+ proto and heme amounts in apple shoots maintained under in vitro culture.
Overall, I found this manuscript interesting and well written. Notwithstanding the complexity of the story due to the very high amount of data presented, the results are concise, clearly described and discussed accordingly.
I have only a list of minor concerns following detailed in the report.
Introduction
This section is well written and provides exhaustive information on the current state of the art contextualizing the authors’ study. The objective of the work is also clearly stated. I have only a few minor comments:
l.74: The proper citation here should be the number 21 of the reference list: Bienert, G.P.; Chaumont, F. Aquaporin-facilitated transmembrane diffusion of 628 hydrogen peroxide. Biochimica et Biophysica Acta 2014, 1840, 1596–1604.
l. 75: The reference here should be the number 25: Dietz, K.J.; Turkan, I.; Krieger, K.A. Redox and reactive oxygen species639 dependent signaling into and out of the photosynthesizing chloroplast. Plant Physiology 2016, 171, 1541-155.
l. 79: Please, change DO to DOES.
l. 80: remove CHANGE.
l. 106: Please, change HAS to HAVE.
l. 120: Please, change ANOTHER to OTHER.
Results & Discussion
l. 167: Considered that the error bars represent standard deviation and that differences are statistically significant for both the GSH content and the GSH/GSSG ratio in the 07-18-094 hybrid samples, variations in the GSH+GSSG concentrations of juvenile and adult samples from the 07-07-119 hybrid should be significant too.
l. 173 Figure 2: I suggest including upper case letter to refer to the single graphs in the figure 2 panel, as done for Figure 1. Accordingly, this should make easier for the reader to immediately visualize the described results.
l. 185: I would also add that the NADP+/NADPH (that should be what displays in Fig. 3D) did not significantly change in the tested samples.
l.189 Figure 3: See the comment made for Figure 2.
l. 197 reference to Fig. 4F: If it is possible, I would move this graph to 4A or describe the results related to fig. 4A-E before this.
l. 203: See the comment previously made for Fig. 4F at line 197.
l. 203: However
l. 239: correlate
l. 267: negatively
l. 310: I would change the part of the sentence from ‘was to control’ to ‘decreased significantly’.
l. 320 Figure 8: ...and of MdHY5, MdLHCB2.4 and MdZTL.
l. 333: WERE consistent
l. 343-344: If I well interpreted results in Fig. 9C, also the expression of GUN6-1 and GUN 6-2 was significantly strongly induced in MdGGT1-RNAi lines, and it was even twice higher than that of GUN5-1 and GUN5-2.
l. 373-374: may BE attributed
l. 375-376: This sentence sounds incomplete.
Methods
l. 437: Given the numerous analyses made, I suggest specifying for which analysis these samples were used.
l. 441-444: Does the M26 rootstock confer any physiological or phenotypical feature to the grafted scion? Is a hybrid genotype as well? Is there any specific reason supporting the choice of M26 as the rootstock?
l. 464: Please, change WERE to WAS.
l. 464-468: Provide information about the time of the day in which the measurement of assimilation rates was conducted. How many leaves were measured for each plant?
l. 513: Did the authors mean low molecular weight RNA here?
l. 513: Please, indicate from which material the RNA samples were extracted.
l. 520: Three hybrids are mentioned at line 436 and it is stated there that samples were collected from all them (l. 437). Hence, why did the authors focus on this hybrid exclusively?
l. 520-524: Information concerning library preparation, sequencing procedure and bioinformatic elaboration of transcriptomic data should be included here.
l. 524-526: As the authors analyze the expression of some candidate genes by RTqPCR (e.g. Fig. 4B & F and Fig. 5 B & C), I would insert this information here.
l. 527: I recommend indicating within this paragraph - or alternatively at the end of each paragraph describing the specific analyses - how many biological replicates (and eventually technical replicates as well) were used for the experiments
Finally, I suggest reducing the number of citations. I guess that almost 100 references are too many for a research, rather than a review, article.
Author Response
Dear Reviewer,
We thank you for your thoughtful suggestions and insights, which have enriched the manuscript and produced a more balanced and better account of the research.
According to your suggestions, we have made modifications in the manuscript by the "Track Changes" function in Microsoft Word. The details are as follows:
Comment 1: l.74: The proper citation here should be the number 21 of the reference list: Bienert, G.P.; Chaumont, F. Aquaporin-facilitated transmembrane diffusion of 628 hydrogen peroxide. Biochimica
Authors’ response: Revised accordingly. At that sentence, we’d better cite two references, one for movement by diffusion and the other for migration via aquaporin.
Comment 2:l.75: The reference here should be the number 25: Dietz, K.J.; Turkan, I.; Krieger, K.A. Redox and reactive oxygen species639 dependent signaling into and out of the photosynthesizing chloroplast. Plant Physiology 2016, 171, 1541-155.
Authors’ response: The statement describes the fact that H2O2 accumulated in chloroplast specifically during adult phase of ontogenesis, without dispersion outwards. This result was reported by our lab years ago. So the citation should be Jia et al., 2017, which is formerly [16] and now it's [11].
Comment 3:l. 79: Please, change DO to DOES.
Authors’ response: Revised accordingly.
Comment 4:l. 80: remove CHANGE.
Authors’ response: ‘vegetative phase change’ is a biological term describing an ontogenetic process. However, we understand the intension of the reviewer should be to avoid the repetitive appearance of ‘change’, so we replaced ‘change’ with ‘transition’.
Comment 5:l. 106: Please, change HAS to HAVE.
Authors’ response: Revised accordingly.
Comment 6:l. 120: Please, change ANOTHER to OTHER.
Authors’ response: Revised accordingly.
Results & Discussion
Comment 7:l. 167: Considered that the error bars represent standard deviation and that differences are statistically significant for both the GSH content and the GSH/GSSG ratio in the 07-18-094 hybrid samples, variations in the GSH+GSSG concentrations of juvenile and adult samples from the 07-07-119 hybrid should be significant too.
Authors’ response: We appreciated this comment and revised accordingly. The significance threshold was set as p < 0.05, and the p value of GSH+GSSG levels between adult and juvenile samples was 0.07, which difference was not statistically significant. However, according to the reviewer’s comment, this difference should not be ignored and we re-organized the interpretation of these data in the text by presenting the p value=0.07.
Comment 8:l. 173 Figure 2: I suggest including upper case letter to refer to the single graphs in the figure 2 panel, as done for Figure 1. Accordingly, this should make easier for the reader to immediately visualize the described results.
Authors’ response: Revised accordingly. We added the capital A B C D to refer to the panels in the Figure 2.
Comment 9:l. 185: I would also add that the NADP+/NADPH (that should be what displays in Fig. 3D) did not significantly change in the tested samples.
Authors’ response: According to the comment of the reviewer, we supplemented the interpretation of NADP+/NADPH ratio results in text.
Comment 10:l.189 Figure 3: See the comment made for Figure 2.
Authors’ response: Revised accordingly. The panels were labelled.
Comment 11:l. 197 reference to Fig. 4F: If it is possible, I would move this graph to 4A or describe the results related to fig. 4A-E before this.
Authors’ response: To make the manuscript easier to read, we followed the reviewer’s suggestion and moved Figure 4F to 4A, by doing this, the labelling of the other panels were successively changed.
Comment 12:l. 203: See the comment previously made for Fig. 4F at line 197.
Authors’ response: Revised accordingly. The same revision as above was done for Figure 4C,D and E. After the position of the Figure 4E changes, the text reference and the chart description are co-modified.
Comment 13:l. 203: However
Authors’ response: Revised accordingly.
Comment 14:l. 239: correlate
Authors’ response: Revised accordingly.
Comment 15:l. 267: negatively
Authors’ response: That sentence has been rephrased according to the comment from the Reviewer 1, hence this error is not applicable now.
Comment 16:l. 310: I would change the part of the sentence from ‘was to control’ to ‘decreased significantly’.
Authors’ response: Revised accordingly. The statement was rephrased.
Comment 17:l. 320 Figure 8: ...and of MdHY5, MdLHCB2.4 and MdZTL.
Authors’ response: We supplemented the missing information in the figure caption, in addition to the gene names, we also defined the genes as Mg-ProtoIX related nuclear genes to make clarity.
Comment 18:l. 333: WERE consistent
Authors’ response: Revised accordingly.
Comment 19:l. 343-344: If I well interpreted results in Fig. 9C, also the expression of GUN6-1 and GUN 6-2 was significantly strongly induced in MdGGT1-RNAi lines, and it was even twice higher than that of GUN5-1 and GUN5-2.
Authors’ response: We appreciate the comment and revised the manuscript accordingly.The result that only the expression of MdGUN5-1 and MdGUN5-2 was remarkably increased in MdGGT1-RNAi lines is correct. But the legend of GUN5 and GUN6 in figure 9C is marked upside down. According to the reviewer’s comment, We modified figure 9C and re-organized the interpretation of these data in the text.
Comment 20:l. 373-374: may BE attributed
Authors’ response: Revised accordingly.
Comment 21:l. 375-376: This sentence sounds incomplete.
Authors’ response: It’s right, the last piece of sentence should be a subordinate clause. We replaced the full stop mark with a comma before that piece.
Methods
Comment 22:l. 437: Given the numerous analyses made, I suggest specifying for which analysis these samples were used.
Authors’ response: We accept the reviewer’s suggestion and made revisions in the 4.1. Plant materials section. We supplemented in the first paragraph of section 4.1. Plant materials that three hybrid trees were used to measure the gene expression and the contents of tetrapyrrole. We described in the next paragraph of section 4.1. Plant materials that the scions grafted on M26 rootstock were used to measure the photosynthetic rate, chlorophyll fluorescence, ROS content, glutathione and NADPH related indicators.
Comment 23:l. 441-444: Does the M26 rootstock confer any physiological or phenotypical feature to the grafted scion? Is a hybrid genotype as well? Is there any specific reason supporting the choice of M26 as the rootstock?
Authors’ response: M26 is a clonal semi-dwarfing rootstock used in apple production and the plant materials can be easily obtained. M26 is certainly a heterozygous genotype like almost all the cultivars of fruit crops, and M26 is propagated by stooling, leafy cutting and other vegetative propagation techniques, so it exhibited high degree of homogeneous among individuals. These are the two reasons we chose M26 as the rootstock to ensure the uniformity of the plants. M26 is a semi-dwarfing rootstock and conferring dwarfing tree architecture and precocious flowering upon the scions grafted on it, but there are no effects on juvenility and other ontogenetic properties.
In an own-rooted hybrid tree, the basal part of the tree is permanently juvenile, and the material on the canopy is in adult reproductive mature phase. We know that the canopy receives much sunlight exposure and the basal suckers obtain less sunlight. We used the scions of hybrids to be grafted onto M26 rootstock just to exclude the influences of different positions and different light exposure.
Comment 24:l. 464: Please, change WERE to WAS.
Authors’ response: Revised accordingly.
Comment 25:l. 464-468: Provide information about the time of the day in which the measurement of assimilation rates was conducted. How many leaves were measured for each plant?
Authors’ response: Revised accordingly. We supplemented the detailed information of time and number of samples for photosynthetic rate and chlorophyll fluorescence measurements.
Comment 26:l. 513: Did the authors mean low molecular weight RNA here?
Authors’ response: Yes. MicroRNAs (miRNAs) are small single-stranded RNAs of about 21 to 23 nucleotides in length, it's a kind of low molecular weight RNA. In most publications, this kind of RNA is usually called microRNA and abbreviated to miRNA.
Comment 27:l. 513: Please, indicate from which material the RNA samples were extracted.
Authors’ response: Revised accordingly.
Comment 28:l. 520: Three hybrids are mentioned at line 436 and it is stated there that samples were collected from all them (l. 437). Hence, why did the authors focus on this hybrid exclusively?
Authors’ response: Transcriptomic analysis is relative expensive because we carried out this experiment by using 18 samples, ie each 30 internode along the cane as a sample, from 1 to 180 internode, with three replicates. So we used one hybrid, 07-07-115, for transcriptomic analysis and the differentially expressed genes of interest were validated by qRT-PCR. Considering the reviewer’s criticism, we supplemented this fact in the first paragraph of 4.1 Plant materials.
Comment 29:l. 520-524: Information concerning library preparation, sequencing procedure and bioinformatic elaboration of transcriptomic data should be included here.
Authors’ response: Revised accordingly. The technical details were briefly described and the reference was cited. The construction of library, sequencing and bioinformatic analysis were completed by Beijing Biomarker Biotechnology co., LTD.
Comment 30:l. 524-526: As the authors analyze the expression of some candidate genes by RTqPCR (e.g. Fig. 4B & F and Fig. 5 B & C), I would insert this information here.
Authors’ response: Revised accordingly. We supplemented some experimental details of RT-qPCR in the text, but these descriptions were inserted to the paragraph before this one. In order to be connective with RNA extraction and primer design.
Comment 31:l. 527: I recommend indicating within this paragraph - or alternatively at the end of each paragraph describing the specific analyses - how many biological replicates (and eventually technical replicates as well) were used for the experiments
Authors’ response: Revised accordingly. We supplemented the information of experimental design at the beginning of the section 4.11. Statistical analysis.
Comment 32:Finally, I suggest reducing the number of citations. I guess that almost 100 references are too many for a research, rather than a review, article.
Authors’ response: Revised accordingly. Several references were removed from the list and the citations of that were deleted. By doing these, the total number of reference was 70, and they were re-numbered in the revised manuscript.
We’re looking forward to your reply.
Sincerely,
Qingbo Zheng